# Monoclonal Human Antibodies That Recognise the Exposed N and C Terminal Regions of the Often-Overlooked SARS-CoV-2 ORF3a Transmembrane Protein

**DOI:** 10.3390/v13112201

**Published:** 2021-11-02

**Authors:** Tyng Hwey Tan, Elizabeth Patton, Carol A. Munro, Dora E. Corzo-Leon, Andrew J. Porter, Soumya Palliyil

**Affiliations:** 1Scottish Biologics Facility, Institute of Medical Sciences, School of Medicine, Medical Sciences and Nutrition, University of Aberdeen, Foresterhill, Aberdeen AB25 2ZP, UK; r01tt19@abdn.ac.uk (T.H.T.); patton.r.elizabeth@gmail.com (E.P.); 2Aberdeen Fungal Group, Institute of Medical Sciences, School of Medicine, Medical Sciences and Nutrition, University of Aberdeen, Foresterhill, Aberdeen AB25 2ZD, UK; c.a.munro@abdn.ac.uk (C.A.M.); dora.corzoleon@abdn.ac.uk (D.E.C.-L.)

**Keywords:** SARS-CoV-2, ORF3a, viroporin, recombinant antibodies, anti-ORF3a mAbs

## Abstract

ORF3a has been identified as a viroporin of SARS-CoV-2 and is known to be involved in various pathophysiological activities including disturbance of cellular calcium homeostasis, inflammasome activation, apoptosis induction and disruption of autophagy. ORF3a-targeting antibodies may specifically and favorably modulate these viroporin-dependent pathological activities. However, suitable viroporin-targeting antibodies are difficult to generate because of the well-recognized technical challenge associated with isolating antibodies to complex transmembrane proteins. Here we exploited a naïve human single chain antibody phage display library, to isolate binders against carefully chosen ORF3a recombinant epitopes located towards the extracellular N terminal and cytosolic C terminal domains of the protein using peptide antigens. These binders were subjected to further characterization using enzyme-linked immunosorbent assays and surface plasmon resonance analysis to assess their binding affinities to the target epitopes. Binding to full-length ORF3a protein was evaluated by western blot and fluorescent microscopy using ORF3a transfected cells and SARS-CoV-2 infected cells. Co-localization analysis was also performed to evaluate the “pairing potential” of the selected binders as possible alternative diagnostic or prognostic biomarkers for COVID-19 infections. Both ORF3a N and C termini, epitope-specific monoclonal antibodies were identified in our study. Whilst the linear nature of peptides might not always represent their native conformations in the context of full protein, with carefully designed selection protocols, we have been successful in isolating anti-ORF3a binders capable of recognising regions of the transmembrane protein that are exposed either on the “inside” or “outside” of the infected cell. Their therapeutic potential will be discussed.

## 1. Introduction

The ongoing, severe, acute respiratory syndrome coronavirus 2 (SARS-CoV-2) pandemic has become an unprecedented challenge to both healthcare systems and economies across the world with varied fatality rates between countries, but all worryingly high [1,2]. This readily transmissible virus [1,2] has attracted substantial research funding to help better understand, diagnose, manage, and treat the disease. Much of this research effort has focussed on the so-called viral “spike” protein (receptor binding domain, RBD) and its interaction, through the ACE2 receptor, with human cells. Blocking of this infection event has been central to the development of both vaccines and biologic therapies [3,4]. Despite being the largest accessory protein of SARS-CoV-2 [5,6], ORF3a has received relatively little research attention, particularly during the pandemic, even though protein homologues were first discovered in SARS-CoV-1 nearly 20 years ago [7,8,9,10]. SARS-CoV-2 ORF3a, characterized as a viroporin [11], has been involved in various pathophysiological activities such as disturbance of cellular homeostasis [11], NLRP3 inflammasome activation [12,13], apoptosis induction [13,14] and disruption of autophagy [15,16,17,18]. 

Phage display technology provides a robust and well-validated antibody discovery platform and has been used successfully for the isolation of several neutralising SARS-CoV-2 anti-RBD antibodies [19,20,21,22,23,24]. Anti-viral, antibody-based therapy has well-recognized advantages including high specificity and affinity, long serum half-life and desirable effector functions [25]. In contrast to readily accessible proteins such as RBD, there remain significant technical challenges when generating antibodies against complex transmembrane, ion channel proteins such as viroporins; not least the difficulty of expressing (or purifying) sufficient target protein in its native conformation, as well as identifying accessible epitopes for antibody binding [26,27,28]. Targeting and modulating the pathological activity of viroporins or eradicating those cells expressing virion proteins may have unexplored therapeutic benefits [27,29,30] including the potential to dampen many of the inflammatory responses that are now known to underpin aspects of long-COVID [31,32,33,34,35,36,37,38]. Here we describe the isolation and characterization of anti-ORF3a human antibodies capable of recognising different epitopes of this challenging transmembrane protein that are antibody-accessible both on the “inside” and “outside” of the infected cell. 

## 2. Materials and Methods

### 2.1. ORF3a Peptides, Plasmid, Virus and Cell Lines

A SARS-CoV-2 ORF3a N-terminal synthetic peptide, N3a (amino acid sequence MDLFMRIFTIGTVTLKQGEIKDATPSDFVRATAT representing positions 1–34 of ORF3a) was custom synthesized (Proimmune, Oxford, UK). 3aC, an ORF3a C-terminal recombinant peptide (amino acid sequence RLWLCWKCRSKNPLLYDANYFLCWHTNCYDYCIPYNSVTSSIVITSGDGTTSPISEHDYQIGYTEKWESGVKDCVVLHSYFTSDYYQLYSTQLSTDTGVEHVTFFIYNKIVDEPEEHVQIHTIDGSSGVVNPVMEPIYDEPTTTTSVPL representing positions 126–275 of ORF3a) was expressed and purified from *Escherichia coli* (NCP0026P, Bioworld Technology, St. Louis, MO, USA). A mammalian expression vector for SARS-CoV-2 ORF3a (with polyhistidine-tag at C-terminal) was sourced from Addgene (Catalogue # 152636, Addgene) and used for transfecting mammalian cell lines for expression of ORF3a. Cell lines used for this study include- Human embryonic kidney cells (HEK293T), African Green monkey kidney epithelial cells (Vero E6) and African Green monkey kidney fibroblast cells (COS-7). The cells were maintained in DMEM supplemented with 10% heat-inactivated fetal bovine serum (FBS), penicillin (100 units/mL), and streptomycin (100 μg/mL) at 37 °C (5% CO_2_). SARS-CoV-2 England/2/2020, supplied by Public Health England, was used for Vero E6 cells infection under Biological Containment Level 3 (BCL3) safety conditions. 

### 2.2. Biopanning and Screening of Antigen Binding Clones Using Phage Display Platform

N3a and 3aC peptides were subjected to biopanning using naïve human phage display antibody libraries as described previously [39,40]. Solution phase biopanning was performed with N3a peptide while 3aC was subjected to both solid and solution phase panning. In solution phase, biotinylated N3a or 3aC peptides were captured by streptavidin coated beads (Dynabeads M-280 11205D, Thermo Fisher, Waltham, MA, USA) and M13KO7 rescued phage particles displaying antibody fragments were co-incubated for target binding. In solid phase biopanning 3aC peptide was directly coated on Nunc MaxiSorp^TM^ plates (44-2404-21, Thermo Scientific, Waltham, MA, USA) and phage particles allowed to bind. Target-bound bacteriophages were eluted with 100 mM Triethylamine and amplified by infecting *Escherichia coli* TG1 cells. Repeated rounds of selection (up to 4 rounds) with progressively decreasing peptide concentrations, was performed to increase the stringency of selection and encourage the enrichment of high affinity ORF3a peptide binders. 

Individual bacterial colonies randomly selected from biopanning were subjected to monoclonal phage ELISA based screening for positive binders by following previously published methods [39,40]. Briefly, MaxiSorp plates were pre-coated with streptavidin prior to adding 1 μg/mL of the respective ORF3a peptide and rescued monoclonal phage supernatant allowed to bind following incubation for 1 h at room temperature. Binding was detected using horseradish-peroxidase (HRP) conjugated anti-M13 antibody (1:5000) (11973-MM05T-H, SinoBiological, Beijing, China) and absorbance values measured at OD450 nm (PerkinElmer Envision 2104 microplate reader). 

### 2.3. Engineering of Single-Chain Antibody Fragment (scAb) and Expression in Bacterial Cells 

Plasmid DNA of peptide binding phage clones were pooled together and their single chain Fv (scFv) genes isolated by restriction digestion using NotI-HF and NcoI-HF (R3189S and R3193S, New England Biolabs, Ipswich, MA, USA) and cloned into the bacterial expression vector pIMS147 [41]. The ligated DNA was ethanol precipitated and used to transform electrocompetent TG1 cells (605021, Lucigen, Middleton, WI, USA). Individual colonies were selected, and DNA sequencing of plasmids confirmed successful conversion of all positive phage clones into the scAb format. 

Selected scAb clones were grown in TB medium and induced with 1 mM Isopropyl β-D-1-thiogalactopyranoside (IPTG) for protein expression in bacterial periplasm [41]. Periplasmic extracts containing histidine-tagged scAbs were purified using immobilized metal affinity chromatography (IMAC) with Ni Sepharose resin (17371202, Cytiva, Marlborough, MA, USA). The concentration of purified scAbs was measured using Pierce™ BCA Protein Assay Kit (23225, Thermo Scientific, Waltham, MA, USA) as per manufacturer’s protocol. Sodium dodecyl sulphate–polyacrylamide gel electrophoresis (SDS-PAGE) was performed using NuPAGE™ 4 to 12%, Bis-Tris protein gel (NP0321PK2, Thermo Scientific, Waltham, MA, USA) to assess scAb purity. 

### 2.4. IgG Conversion of Selected ORF3a scAbs 

N3a peptide binding scAb, N3aB02, was reformatted into a mouse monoclonal antibody by cloning its variable heavy (VH) and variable light (Vk) genes into a dual plasmid eukaryotic vector system encoding the constant domain genes of mouse IgG2a isotype and murine kappa chain respectively. Human Embryonic Kidney cells (HEK293-F) (Life Technologies, Carlsbad, CA, USA) were transfected with plasmids harbouring reformatted antibody heavy and light chain genes using polyethyleneimine (PEI). Transfection was carried out using 1 mg of total DNA (500 μg each of VH and Vκ plasmid DNA) and 1000 mL of cultured HEK293-F cell suspension maintained in sterile Freestyle 293 expression medium (Invitrogen) without antibiotics at 37 °C, with 8% CO_2_, 125 rpm shaking. The transfected cells were allowed to express N3aB02 mAb for a week before the supernatant was collected and purified using protein A beads (ProSep^®^ Ultra Plus, Merck Millipore, Burlington, MA, USA). N3aB02 mAb was eluted using 100 mM glycine (pH 3.0), dialysed against 1× PBS and quantified as above.

### 2.5. ORF3a Transfection and Virus Infection

ORF3a plasmid, confirmed by sequencing using CMV Fwd primer (DNA Sequencing Services, University of Dundee) was used to transfect Vero E6 and COS-7 cells grown in DMEM, high glucose, GlutaMAX™ Supplement, pyruvate medium (31966021, Gibco™) with 1% (*v*/*v*) Penicillin-Streptomycin and 10% (*v*/*v*) FBS at 37 °C in a 5% CO_2_ humidified incubator. Transient transfection for heterologous expression of ORF3a was performed using lipofectamine™ 3000 transfection reagent (L3000015, Invitrogen, Carlsbad, CA, USA) as per manufacturer’s protocol. The transfected cells were fixed with 2% paraformaldehyde for 15 min. For virus infection, Vero E6 cells were co-incubated with SARS-CoV-2 for 48 h prior to fixation with 4% paraformaldehyde for 3 h with all procedures carried out at BCL3. 

### 2.6. Western Blot Using ORF3a Transfected Cell Lysate 

ORF3a transfected and non-transfected HEK293T cells were lysed using RIPA Buffer (Pierce^TM^ 89900, Thermo Scientific, Waltham, MA, USA) in protease inhibitor cocktail (Halt^TM^ 1860932, Thermo Scientific, Waltham, MA, USA) as per manufacturer’s protocol. SDS-PAGE was conducted using 12 μg of each cell lysate and for western blotting, proteins were transferred to Immune-blot PVDF membrane (162-0219, BioRad, Hercules, CA, USA), blocked with 5% Marvel PBS prior to incubation with 1 μg/mL of respective scAb in 5% MPBS for 1 h at room temperature. Following washing, the membranes were incubated with respective HRP-conjugated secondary antibodies and signals generated using Clarity western ECL substrate (1705061, BioRad, Hercules, CA, USA). ORF3a polyclonal antibody (NCP0037, Bioworld Technology, St. Louis, MO, USA) and anti-polyhistidine HRP monoclonal antibody (A7058, Sigma Aldrich, St. Louis, MO, USA) were used as positive controls. 

### 2.7. Immunofluorescence Microscopy

The fixed transfected cells were stained using 5 μg/mL Alexa Fluor 488 (AF488) conjugated N3aB02 or Alexa Fluor 647 (AF647) conjugated 3aCA03 scAbs for 1 h at room temperature. Staining was performed without permeabilization or permeabilized with 0.5% saponin in PBS for 15 min at room temperature. After washing, the cells were mounted in ProLong™ glass antifade mountant (P36984, Invitrogen) and assessed using confocal microscope LSM880. Colocalization was performed using the colocalization tool in Zen Black (Zeiss, Oberkochen, Germany).

Fixed SARS-CoV-2 virus infected cells were permeabilized as above, stained with N3aB02 or 3aCA03 scAbs for 1 h at room temperature followed by FITC labelled anti-Human Kappa Light Chain (F3761, Sigma-Aldrich, St. Louis, MO, USA) as secondary antibody. The cells were examined using Ultraview VoX spinning disk confocal microscope as per local standard operating procedure for COVID-19 specimens. 

### 2.8. Surface Plasmon Resonance (SPR) Kinetic Analysis

N3aB02 and 3aCA03 antibodies binding kinetics were analysed using surface plasmon resonance (Biacore X100) with a biotin CAPture kit (28920233, Cytiva, Marlborough, MA, USA). Biotinylated N3a or 3aC ligand was immobilized on the sensor surface and several concentrations of test antibodies (N3aB02 scAb, N3aB02 mAb or 3aCA03 scAb) were passed over as analytes. This allowed real-time detection of N3a or 3aC ligand and antibody (or fragment) interaction profile which was recorded as a sensorgram. The sensorgrams were analysed using Biacore X100 evaluation software (version 2.0.2) with both blank cycle and reference flow cell subtraction. The affinity constant (*K*_D_—equilibrium dissociation constant) is calculated as the ratio of association and dissociation rate constants (*K*_D_ = *k*_d_/*k*_a_).

## 3. Results

### 3.1. Isolation of ORF3a Specific Human Recombinant Antibodies

The scFv phage display library was subjected to multiple rounds of stringent selection using ORF3a peptides and positive clones were identified from phage monoclonal ELISA signals. Plasmid sequencing confirmed unique, full-length scFv genes and twelve 3aC and eight N3a clones were selected for scAb reformatting and further characterization. 

The antigen binding activity of soluble scAbs produced in a bacterial expression system was confirmed by ORF3a peptide binding ELISA. Interestingly, only N3aB02 and 3aCA03 scAbs bound to the full length ORF3a (MW ~35 kDa) in a transfected cell lysate western blot (Figure 1A). The EC50 values (half maximal binding response) of N3aB02 and 3aCA03 scAbs to their peptide antigens were estimated at a respectable 0.58 μg/mL and 0.31 μg/mL respectively (Figure 1B). 

### 3.2. N3aB02 and 3aCA03 Antibodies Bind Full Length ORF3a in Transfected and SARS-CoV-2 Infected Cells 

The ORF3a N terminus is predicted to be protruding extracellularly, while the C terminus is retained within the cytoplasm [11]. Fluorescence microscopy revealed that the N3aB02 scAb binds to ORF3a transfected cells, either permeabilized or non-permeabilized, confirming the extracellular location of the antibody binding region (Figure 1C). 3aCA03 scAb binding was confined to the intracellular domain of ORF3a since the staining was only observed when the cells were permeabilized (Figure 1C). Both N3aB02 and 3aCA03 scAbs bound to SARS-CoV-2 infected Vero E6 cells (Figure 1D).

### 3.3. N3aB02 Colocalized with 3aCA03 scAb upon Binding to ORF3a

Fluorescent images suggested co-localization of AF488-tagged N3aB02 and AF647-tagged 3aCA03 scAbs when they were incubated simultaneously with permeabilized ORF3a transfected cells (Figure 1E). This phenomenon was demonstrated with a high Manders’ colocalization coefficient (0.97) in both transfected Vero E6 and COS-7 cells (Figure 1F). This also suggested that both N3aB02 and 3aCA03 scAbs could potentially form a diagnostic antibody pair capturing an ORF3a protein at “opposite ends” without any apparent steric hindrance. 

### 3.4. scAb to mAb Conversion Improved Binding Affinity

Using SPR, the binding kinetics and overall affinity of N3aB02 antibody in scAb and IgG mAb formats (monovalent vs. bivalent) to its target peptide was measured. The results confirmed an improvement in binding affinity following mAb conversion, which is associated with a typical avidity effect due to the two binding sites on an IgG molecule compared to one binding site on a scAb. The N3aB02 scAb kinetics data were found to fit better with a two-state reaction model (Figure 1G and Appendix A) compared to a 1:1 Langmuir binding model for N3aB02 mAb (Figure 1G). This may indicate some conformational changes following binding of the N3aB02 scAb to 3a ligand and would need further evaluation. N3aB02 scAb, analysed with a two-state reaction model, generated *k*_a1_, *k*_d1_, *k*_a2_ and *k*_d2_ values of 1.4 × 10^5^ M^−1^s^−1^, 0.04 s^−1^, 0.009 s^−1^ and 0.002 s^−1^, respectively. This resulted in an overall *K*_D_ of 57 nM (Figure 1G). N3aB02 mAb kinetics data, fitted with the Langmuir binding model, produced *k*_a_ of 7.7 × 10^5^ M^−1^s^−1^, *k*_d_ of 0.01 s^−1^ and a calculated *K*_D_ of 16 nM (Figure 1G). 

The association and dissociation rate constants for 3aCA03 scAb based on 1:1 Langmuir binding model was 4.2 × 10^5^ M^−1^s^−1^, 5.4 × 10^−4^ s^−1^, respectively, with an overall *K*_D_ value of 1.3 nM (Appendix A).

## 4. Discussion

The ORF3a protein is believed to be present at various cellular locations, such as the plasma membrane, cytoplasm and endosome of infected host cells [15,42]. When embedded in the plasma membrane, it has six functional domains with extracellular N terminal and cytoplasmic C terminal regions [11,43]. In this study, we have isolated SARS-CoV-2 ORF3a epitope-specific antibodies from a naïve human phage library. Both ORF3a N and C termini peptides were used to isolate these antibodies. Using peptide antigens can overcome the challenges of producing full length protein for biopanning, however the linear nature of peptides does not always represent their true, native conformation. In our study, a panel of ORF3a peptide binders were successfully identified. Amongst these binders, only one in each group (N3aB02 and 3aCA03) was shown to bind to full length ORF3a protein (Figure 1A), recognising the N and C terminus of the protein, respectively. These two clones bound to their epitopes with high affinity in EC_50_ and/or SPR analyses and with enhanced affinity when reformatted from a monovalent antibody fragment (scAb) to a bivalent IgG mAb. 

SARS-CoV-2 ORF3a has been proposed to have a number of intracellular functions including autophagy disruption, caveolin binding and a possible RNA-binding function [43,44]. Inhibition of ORF3a might therefore compromise SARS-CoV-2 pathogenesis. In this study, we observed that following 48 h of SARS-CoV-2 infection, a population of swollen ‘blebbing’ Vero E6 cells took up 3aCA03 scAb with strong binding to these blebs (Appendix A). Microscopy studies would suggest these pyroptotic-like cells, characterized by “blebbing appendages”, are apparently regions containing high densities of ORF3a protein, with anti-ORF3a antibodies binding without a requirement for permeabilization, in contrast to transfected lines (Figure 1C). This might reflect ORF3a involvement in SARS-CoV-2 infection-related programmed cell death. This observation would also fit with the reported involvement of NLRP3 inflammasome activation and pyroptosis in SARS-CoV-2 [45,46,47,48]. The binding pattern of SARS-CoV-2 anti-ORF3a binders was in stark contrast to that seen for a commercially sourced anti-RBD antibody (Appendix A) which showed a much more diffused binding signal from predominantly inside and throughout the infected cell. Additional studies would be required to confirm this hypothesis but based on the evidence to date, from ourselves and others, this does provide a credible explanation for the binding pattern observed.

While vaccination has been the central strategy in preventing or dampening the viral pandemic [49], vaccination is clearly less effective in certain circumstances exacerbated by the emergence of new SARS-CoV-2 variants which may limit or shorten the impact of certain vaccine therapies [50,51,52]. In addition, there remains an important sub-set of individuals, typically with compromized immune systems, that respond less well to vaccines [53,54,55], thus necessitating the development of additional antiviral strategies. In those regions of the world claiming a level of pandemic-control, a new and growing epidemic of patients suffering from longer-term post-infection complications, often referred to by the catch-all “long-COVID”, is being increasingly reported. Whilst long-COVID is a condition that is still not well understood, there are some pointers in the general condition that suggest that inflammation responses, linked to long term viral infection, may be at its core, with various tissues and even different organs affected [37,56,57,58,59]. 

N3aB02, as a fully human monoclonal antibody, targeting the extracellular domain of ORF3a, might be a potential candidate for depleting or modulating the SARS-CoV-2 infected cell populations and the resulting inflammation they may cause. A previous study has shown the ability of the related SARS-CoV-1 ORF3a N terminal targeting antibodies, derived from convalescent patients’ plasma, to induce the destruction of ORF3a-expressing cells by activation of the complement pathway [60]. In addition, Akerstrom et al. (2006) have demonstrated that rabbit antisera specific for the N terminus of SARS-CoV-1 ORF3a (representing amino acids 15–28) resulting in a neutralising effect 48 h post infection in virus infected Vero E6 cells [61]. There was no corresponding neutralizing effect using antisera that recognized the C terminal region of SARS-CoV-1 ORF3a in two separate studies [61,62] after 2–4 days of infection. Previous studies have also revealed a significant proportion of convalescent sera or plasma of patients with ORF3a antibodies compared to patients who had succumbed to SARS-CoV infection [60,63]. This would support further exploration of the therapeutic potential of ORF3a antibodies. 

## 5. Conclusions

These examples cited above show a clear N terminal bias for antibody neutralization of viral infection via the recognition of ORF3a related proteins. In our, albeit limited infection studies (in vitro and not in vivo), once programmed cell death has been triggered by the presence of ORF3a protein within the infected cell, significant membrane disruption is observed. This damage is such that fully human antibodies that recognize either the N or C terminus of the SARS-CoV-2 ORF3 protein are able to bind unhindered and may therefore have utility both as treatments for viral infection and possibly the related condition of long-COVID. 

## Figures and Tables

**Figure 1 viruses-13-02201-f001:**
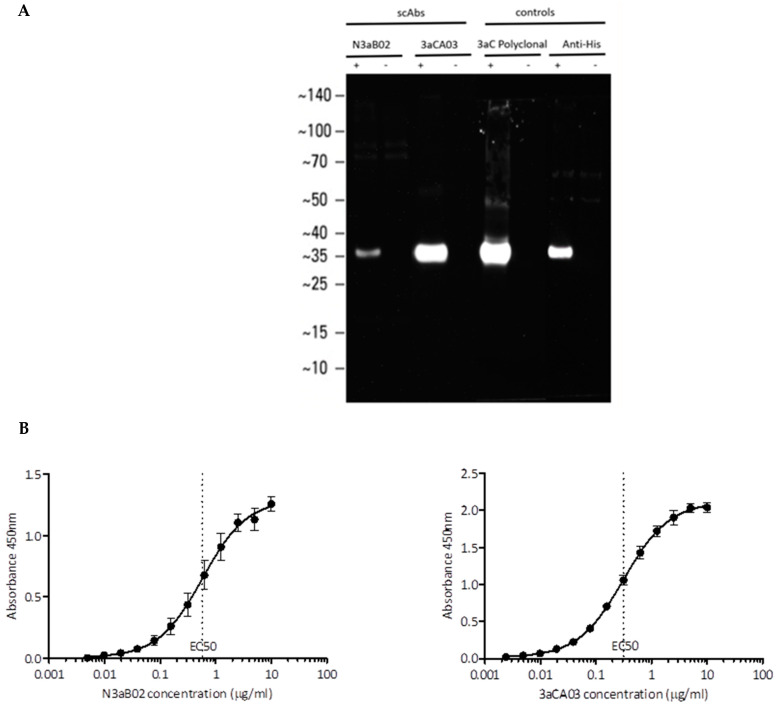
Summary of 3a antibodies characterization. (**A**) Western blot analysis of N3aB02 and 3aCA03 scAbs binding to ORF3a cell lysate (+ ORF3a transfected HEK293T cells; − non-transfected cells). (**B**) EC50 for N3aB02 and 3aCA03 scAbs binding to respective peptide antigens (data presented are mean +/−SEM of three experimental replicates). (**C**) Immunofluorescence (IF) microscopy images showing binding of AF488 conjugated N3aB02 and AF647 conjugated 3aCA03 scAbs to ORF3a transfected cells in permeabilized or non-permeabilized cells (scale bar = 5 μm). (**D**) IF microscopy images showing binding of N3aB02 and 3aCA03 scAbs to SARS-CoV-2 infected cells. FITC labelled anti-Human Kappa light chain antibody was used for fluorescent signal generation (scale bar = 12 μm). (**E**) IF staining showing colocalization of N3aB02-AF488 and 3aCA03-AF647 scAbs in ORF3a-transfected cells (scale bar = 5 μm). (**F**) Co-occurrence scatterplot of AF488 (N3aB02) and AF647 (3aCA03) fluorescent signals representing the degree of overlap of ORF3a N and C terminal antibodies binding in transfected cells. (**G**) SPR sensorgrams of N3aB02 scAb and N3aB02 mAb from multi cycle kinetics. Colored lines represent experimental data with different scab or mAb concentrations and black lines represent fitted curves.

## Data Availability

All data generated in this study could be found in this article.

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
