# Peer review of "Monoclonal Human Antibodies That Recognise the Exposed N and C Terminal Regions of the Often-Overlooked SARS-CoV-2 ORF3a Transmembrane Protein"

_viruses, 2021, doi:10.3390/v13112201_

Round 1

Reviewer 1 Report

The authors aimed to describe the isolation and characterisation of anti-ORF3a human antibodies capable of recognizing different epitopes of this challenging transmembrane  protein that are antibody-accessible both on the “inside” and “outside” of the infected cell.

The study covers some issues that have been overlooked in other similar topics. The structure of the manuscript appears adequate and well divided in the sub-paragraphs. Moreover, the study is easy to follow, but some issues should be improved. The manuscript needs moderate grammar correction. Please also check typos thorough the text.

Discussion section: Limitations of the study should be reported.

Conclusion Section: This paragraph is missing. Please add it.

Author Response

We would like to thank the reviewers for their careful and critical consideration of our manuscript and for providing us with constructive feedback and comments.

Our responses to reviewer#1 are as follows:

Reviewer #1 Comments

The authors aimed to describe the isolation and characterisation of anti-ORF3a human antibodies capable of recognizing different epitopes of this challenging transmembrane protein that are antibody-accessible both on the “inside” and “outside” of the infected cell.

The study covers some issues that have been overlooked in other similar topics. The structure of the manuscript appears adequate and well divided in the sub-paragraphs. Moreover, the study is easy to follow, but some issues should be improved. The manuscript needs moderate grammar correction. Please also check typos thorough the text.

Discussion section: Limitations of the study should be reported.

Conclusion Section: This paragraph is missing. Please add it.

Reply to Reviewer #1

We appreciate the reviewer comment Discussion section: Limitations of the study should be reported and Conclusion Section: This paragraph is missing. Please add it’. We have addressed these points in the revised version of the manuscript.

Response to Specific comments

  1. Discussion section: Limitations of the study should be reported
  2. Conclusion Section: This paragraph is missing. Please add it –

Our response – Study limitation and conclusion included in the revised manuscript, please see lines 347-355

Yours sincerely

TyngHwey Tan et al.

Reviewer 2 Report

In the manuscript by Hwey Tan et al, the authors describe the identification of monoclonal antibodies capable of recognizing epitopes located at the N or C terminal end of the SARS-CoV2 viroprotein ORF3. The authors provide solid and convincing data on antibody specificity of binding of the antibodies  to the ORF3 using different methodologies. This study provides a great scientific contribution toward the development of novel biotherapeutics and antivirals against CoVID19. Please see my comments below:

  1. Although the authors only discussed the biotherapeutic potential of the antibodies identified in this study, I am surprised why they did not show its ability to affect viral replication given their capabilities of conducting infection with SARS-CoV2.
  2. The authors need to provide the epitope sequences of the antibodies identified in this study.
  3. Line 63: “transmembrerane” typo
  4. Figure 2. The authors should be more consistent with the images that are being displayed. Some are displayed with higher magnification and others not. Why the distinction?
  5. Line 279: “cab alization” typo? Do you mean colocalization?

Author Response

Response to Reviewers

Manuscript - Monoclonal human antibodies that recognise the exposed N and C terminal regions of the often-overlooked SARS-CoV-2 ORF3a transmembrane protein by Tan et al.

We would like to thank the reviewers for their careful and critical consideration of our manuscript and for providing us with constructive feedback and comments.

Our responses to reviewer #2 are as follows:

Reviewer #2 Comments

In the manuscript by Hwey Tan et al, the authors describe the identification of monoclonal antibodies capable of recognizing epitopes located at the N or C terminal end of the SARS-CoV2 viroprotein ORF3. The authors provide solid and convincing data on antibody specificity of binding of the antibodies to the ORF3 using different methodologies. This study provides a great scientific contribution toward the development of novel biotherapeutics and antivirals against CoVID19. Please see my comments below:

  1. Although the authors only discussed the biotherapeutic potential of the antibodies identified in this study, I am surprised why they did not show its ability to affect viral replication given their capabilities of conducting infection with SARS-CoV2.
  2. The authors need to provide the epitope sequences of the antibodies identified in this study
  3. Line 63: “transmembrerane” typo –
  4. Figure 2. The authors should be more consistent with the images that are being displayed. Some are displayed with higher magnification and others not. Why the distinction?
  5. Line 279: “cab alization” typo? Do you mean colocalization? –

  Reply to Reviewer #2

We appreciate your positive feedback. Please see below responses to specific comments.

  1. Although the authors only discussed the biotherapeutic potential of the antibodies identified in this study, I am surprised why they did not show its ability to affect viral replication given their capabilities of conducting infection with SARS-CoV2.

Our response - The referee is correct that biotherapeutic potential was discussed rather than demonstrated.  We believe that for this to be properly assessed we will require to carry out in vivo infection studies rather than in vitro studies.  We also think that careful optimisation of in vivo infection conditions may be required to properly identify a positive (or even negative) effect on the progression of the disease.  This is in contrast to those studies with anti-RBD antibodies which can impact directly and immediately on the infection event.  What would be even more interesting is if animal models could be manipulated to mimic long-Covid as it is in this setting that we feel our antibodies may have their most beneficial effects.  We are now working with collaborators to secure this data but it is quite a significant undertaking.

  1. The authors need to provide the epitope sequences of the antibodies identified in this study

Our response –

Epitope – 1 – amino acid sequence added in lines 67-68

Epitope – 2 - amino acid sequence added in lines 70-73

  1. Line 63: “transmembrerane” typo

Our response – Typo corrected

  1. Figure 2. The authors should be more consistent with the images that are being displayed. Some are displayed with higher magnification and others not. Why the distinction?

Our response - Figures changed to uniform magnification in the revised manuscript

  1. Line 279: “cab alization” typo? Do you mean colocalization? –

Our response - It is colocalization and our original manuscript had the correct spelling. We suspect this is a formatting error introduced while the original manuscript was uploaded onto the journal website.

Yours sincerely

TyngHwey Tan et al.

Round 2

Reviewer 1 Report

The authors have addressed to all comments. Thank you.

Reviewer 2 Report

The authors have responded to the questions raised during the review process.